# Exploring Chinese Consumers’ Perception and Potential Acceptance of Cell-Cultured Meat and Plant-Based Meat: A Focus Group Study and Content Analysis

**DOI:** 10.3390/foods14091446

**Published:** 2025-04-22

**Authors:** Muhabaiti Pareti, Junsong Guo, Junjun Yin, Qiankun Liu, Nadire Abudurofu, Abulizi Bulibuli, Maurizio Canavari

**Affiliations:** 1College of Economics and Management, Xinjiang Agricultural University, Nonda East Road 311, Urumqi 830052, China; 19275313559@163.com (J.G.); yjunjun@xjau.edu.cn (J.Y.); 18199861480@163.com (N.A.); 2Department of Agricultural and Food Sciences, Alma Mater Studiorum-University of Bologna, Viale Guiseppe Fanin 50, 40127 Bologna, Italy; qiankun.liu2@unibo.it; 3School of Business and Administration, Xinjiang University of Finance &Economics, Beijing Middle Road 449, Urumqi 830012, China

**Keywords:** plant-based meat, cultured meat, consumer perception, consumer acceptance, focus group interview, content analysis

## Abstract

(1) Background: In recent years, meat alternatives, including plant-based and animal cell-cultured meat, have attracted substantial interest among Chinese food science researchers and consumers, prompting considerable debate; (2) Methods: This study utilizes qualitative research methods, specifically focus group interviews with 59 participants across five administrative regions and seven cities in China, to explore consumer knowledge, perceptions, and potential acceptance of meat substitutes; (3) Results: The findings reveal that Chinese consumers generally exhibit a low level of understanding of new meat substitutes, particularly animal cell-cultured meat. Although participants acknowledge the potential environmental, resource-saving, and animal welfare benefits associated with meat substitutes, they also express concerns about perceived risks, such as artificial taste, high costs, market monopolization, diminished consumer welfare, and adverse impacts on traditional animal husbandry and employment. Despite a willingness to try meat substitutes, the regular purchase and consumption of these remain limited. The acceptance of meat substitutes is influenced by factors including personal characteristics, price, safety, and the authenticity of taste; (4) Conclusions: The study concludes that legislative support, technological advancements in production and regulation, price reductions, and the establishment of a robust traceability system may enhance consumer confidence and acceptance of meat substitutes in China.

## 1. Introduction

With the increasing global demand for meat products, various meat alternatives, particularly plant-based meat and cell-cultured meat, have emerged as prominent options [1]. As alternative protein sources, these substitutes have the potential to address the growing need for meat consumption in the future [2]. Currently, meat substitutes represent a significant topic of interest in food science and production, fueled by ethical and environmental concerns that have sparked substantial debate [3]. Proponents argue that meat substitutes offer a safer alternative to conventional meat, as producers can control both the production environment and the nutritional content [4,5]. Some researchers suggest that these alternatives can mitigate the negative effects associated with antibiotics and reduce contamination risks from bacteria and viruses present in traditional meat production [6]. Supporters emphasize the potential environmental benefits of meat substitutes, which include lower water and land usage and reduced greenhouse gas emissions compared with conventional meat production [7,8]. Moreover, meat substitutes do not involve the mass slaughter of animals, making them a preferred choice for animal conservationists [9], and are often favored by vegetarians and vegans for their ethical considerations [10]. However, some critics express concerns about the impact of meat substitutes on traditional animal husbandry, suggesting potential disruptions to existing agricultural practices [11,12]. Additionally, the complex production processes of meat substitutes may introduce potential health risks, as noted by some scholars [13]. Research also indicates that conventional meat production systems may use less energy, water, and land and produce fewer greenhouse gases than some meat substitute production methods [14,15]. Furthermore, the production of cultured meat still involves animal exploitation, as animals are required for the extraction of stem cells, raising ethical concerns among critics [16].

With the development of China’s economy and increases in consumption levels, Chinese consumers have shifted from a diet primarily based on grains to one that includes a significant amount of meat products. Since the early 1990s, meat consumption in China has increased steadily, establishing the country as the world’s largest meat consumption market [17]. In 2021, Chinese consumers accounted for nearly 100 million tons of meat, representing 27% of global meat consumption. This growing demand for meat products in China may further exacerbate the need for grasslands, expand land use, and increase the reliance on conventional meat systems, which, in turn, pose challenges such as environmental pollution, grassland degradation, and animal welfare concerns [18]. In recent years, plant-based meat substitutes have started to gain traction in the Chinese market. With rising public interest and continuous technological advancements, both researchers and enterprises in China are increasingly investing in the development of meat substitutes. However, compared with the United States and European countries, China still lags in regard to the technology, market share, legislation, and regulatory frameworks related to meat substitutes. Despite their potential, meat substitutes have not yet achieved large-scale commercialization in China.

Given China’s status as the largest meat consumption market globally, understanding Chinese consumers’ perceptions of meat alternatives is essential [19]. In the field of consumer behavior, the Theory of Planned Behavior is usually applied to explore consumers’ behavioral intentions. Within this theory, attitudes, subjective norms, and perceived behavioral control jointly determine behavioral intentions. Since both behavioral attitudes and subjective norms involve an individual’s perception of behaviors and others’ expectations, they can be classified as aspects of perception. Therefore, in numerous studies concerning consumer behavior and acceptance, perception typically precedes acceptance [20,21,22], such as of perceived costs, ethical concerns [23], perceived usefulness, and perceived ease of use [24]. However, some scholars have categorized acceptance under internal factors and perception under external factors; this classification demonstrates a relatively limited degree of progression [25]. The objective of this paper is to collect information on consumer perception and to develop an acceptance model. Consequently, before exploring acceptance, our primary focus should be on consumers’ perception. The current controversy surrounding meat substitutes has led many consumers to adopt a cautious and reserved stance toward their consumption. A survey of 1004 Chinese consumers found that a vast majority were unfamiliar with meat alternatives, including cultured meat; nearly 22% expressed opposition to cultured meat, while approximately 50% remained neutral [26]. It is important to note that previous studies have indicated that bias may arise due to the confounding effects associated with different naming conventions. For instance, “cultivated meat” can be translated into “cultured meat”, “lab-grown meat”, “in vitro meat”, or “cell-based meat”, terms that are less familiar to Chinese consumers compared with “artificial meat” [27].

While some studies have investigated Chinese consumers’ perceptions and potential acceptance of meat substitutes, the majority have relied on quantitative research methods. These studies often quantify the proportion of consumers who accept meat substitutes or use econometric models to analyze the primary factors influencing consumer choices; however, they lack an in-depth exploration of the underlying reasons behind consumers’ attitudes and behaviors. This study seeks to address this gap by exploring Chinese consumers’ knowledge, perceptions, and potential acceptance of meat substitutes through qualitative research methods. Specifically, this study aims to address the following questions:*1.* *What is the level of knowledge among Chinese consumers about meat substitutes?**2.* *What are Chinese consumers’ perception of meat substitutes?**3.* *What is the potential acceptance of meat substitutes among Chinese consumers?*

The examination of consumers’ knowledge, perceptions, and acceptance of meat substitutes is a significant focus within international food marketing research. However, in the Chinese context, particularly concerning cultured meat, the lack of commercialization has led to consumers exhibiting ambivalent, cautious, and skeptical attitudes toward these products. Although qualitative research inherently carries certain limitations, such as limited sample size, subjective participant selection, and potential challenges in generalizing findings, this study provides a foundational basis for future quantitative research. By offering exploratory insights into Chinese consumers’ perspectives and potential acceptance of meat substitutes, this study aims to catalyze more extensive research efforts in this area.

## 2. Materials and Methods

Qualitative research is a foundational approach for exploring and elucidating real-world issues. Previous studies have employed various qualitative methodologies, such as in-depth interviews and focus group discussions, to examine consumer perceptions, attitudes, and preferences [28,29,30]. Among these methods, focus groups and in-depth interviews are particularly powerful tools that are widely used in social science research [31]. Compared with in-depth interviews, focus group interviews offer distinct advantages, such as the ability to quickly gather collective insights from group members and to facilitate dynamic interactions that produce comprehensive and nuanced data [32]. Additionally, the interactive nature of focus group interviews enhances participant engagement, allowing respondents to build on topics raised by their peers, thereby broadening and deepening the scope of discussion [33].

### 2.1. Selection of Participants and Interview Locations

The recruitment process for focus group members was as follows: First, recruitment interview information was released on various social media platforms. For instance, when we planned to conduct focus interviews in Jinan, Shandong Province, recruitment announcements were posted on multiple platforms such as Zhihu, Tieba, and Douyin. Second, a preliminary screening of the interviewees who had registered was carried out. For those who met our interview criteria, we invited them to sign an informed consent commitment. During the formation of interview groups, meticulous arrangements were made to ensure that, as far as possible, different interview groups had different concentrated age brackets. This was to make the interview perspective more comprehensive and to obtain richer and more representative data.

The focus group interviews were conducted between January and March 2024, involving a total of 59 participants, including 32 males and 27 females. In selecting participants for the focus groups, factors such as age, gender, region, and urban or rural residency were carefully considered. Participants from Beijing, Shanghai, Urumqi, and Ili predominantly consisted of young consumers aged 20–30, while those from Linyi and Changsha were aged 30–45 and 40–60, respectively. Notably, all the focus group participants from Linyi were rural residents. While the age ranges of the different groups varied, the inclusion criteria were consistently defined as the following: regular food shoppers with internet access, familiarity with or prior purchase of meat substitutes, and aged 20 years or older (gender-neutral). The inclusion criteria aimed to maintain an optimal group size of 6 to 12 individuals per session to prevent inefficiencies associated with excessively large or small groups. Additionally, efforts were made to recruit participants with similar characteristics to minimize intra-group disparities and to ensure more cohesive discussions.

Due to time and budget constraints, this study conducted seven focus group interviews across seven cities in China. These interviews were strategically distributed across the eastern, central, and western regions to ensure geographic representation from both the northern and southern parts of the country. The selected cities included Beijing, Shanghai, Changsha (Hunan), Jinan (Shandong), Linyi (Shandong), Urumqi (Xinjiang), and Ili (Xinjiang), as detailed in Table 1. Beijing and Shanghai, as major political, economic, and cultural centers of China, have residents who generally possess higher levels of education and who exhibit a greater acceptance of and greater purchasing power for new products compared with people from other regions. Shandong Province, a developed coastal area with a robust agricultural product processing industry, is also a key region for the meat substitute industry in China, with development levels approaching those of Beijing and Shanghai. Changsha, located in Hunan Province in south–central China, is notable for its residents’ strong preference for spicy food, with stinky tofu being a prominent local dish. Given that tofu is a primary source of plant-based protein for meat substitutes, it was hypothesized that residents of Hunan would demonstrate a higher tolerance for the taste of meat substitutes, making Changsha a suitable location for this study. Urumqi and Ili, situated in Xinjiang, an administrative region in northwest China, represent areas with a strong tradition of animal husbandry. Consumers in these regions tend to have a preference for meat products and generally possess lower purchasing power compared with those in the more developed eastern coastal regions. Consequently, these Xinjiang locations were included as representative inland areas for the focus group interviews.

Due to considerations of survey convenience and budget constraints, focus group interviews for the two mega cities, Beijing and Shanghai, were conducted online, whereas face-to-face interviews were conducted in the other five regions. Each focus group discussion lasted approximately 90 min.

### 2.2. Data Collection

Prior to initiating the focus group interviews, the researchers ensured that all participants were thoroughly briefed on the research objectives and content. Consent was obtained from each participant, who signed an informed consent form. Additionally, participants engaged in the discussions anonymously to maintain privacy. At the beginning of each session, the facilitator outlined the primary discussion topics and established the rules for the focus group. Participants were instructed to focus on plant-based meat and animal cell-cultured meat. The interview was structured into three distinct sections: consumer knowledge, perception, and acceptance of meat substitutes. In the initial phase, participants were asked to describe their understanding of conventional meat products and their substitutes. The discussion then shifted to explore the participants’ perceptions of meat substitutes. In the final phase, the participants deliberated on their willingness to accept these alternatives.

Each focus group session was conducted under the supervision of two researchers: one facilitated and moderated the discussion, while the other recorded relevant participant responses. Rigorous quality control measures were employed, including the repeated review and verification of interview transcripts by the researchers. Subsequently, all textual data were imported into the NVivo 12 software for systematic coding and content analysis. The Ucinet 6.365 software was then utilized for co-occurrence analysis to explore and visualize relationships among words and to construct deep-level associations and structures between nodes. The focus group interviews were conducted according to a predetermined set of questions, as outlined in Table 2.

### 2.3. Data Analysis

We carried out three-stage coding for the texts of the focus interviews. First, after converting the interview recordings into text files, we started conducting open-coding on these complex and disordered texts. During this process, we always maintained an open and inclusive attitude, carefully decomposing the massive amount of collected data, meticulously extracting valuable information units from them, and attaching corresponding conceptual labels to them; for example, when a consumer mentioned, “I’ve heard that artificial meat is good for the environment, so I’m a bit interested”. Based on this, we extracted the initial code “environmental benefit”. Then, came the axial-coding stage. We took the concepts and categories generated by the open-coding as the foundation and re-examined and sorted the data with caution. Through this method, the originally scattered categories can be closely connected according to specific logical relationships, thereby further delving into the connotation of the data. For instance, “environmental benefit” and “pollution cognition” are grouped into one category and named “environment”, making the connection among these pieces of information clearer. Finally, we utilized selective coding. In this process, we accurately determined a core category from the numerous categories obtained through the analysis, and then systematically established associations between the remaining categories and this core category in an orderly manner. In this way, an integrated and coherent theoretical system was constructed, laying a solid theoretical foundation for our research and making it more scientific and convincing. Following the analysis of the seven focus groups, data saturation was deemed to have been attained. That is, no novel themes could be discerned from the remaining groups, which suggests that the collected data encapsulated the diverse viewpoints of the research sample regarding the potential acceptance of plant-based meat and cell-cultured meat. In qualitative research, the researcher’s position is crucial, as varying stances can introduce biases into the study conclusions. To mitigate such differences, our study employed a two-step approach: researchers initially coded the data independently and subsequently convened for discussions to align themselves on the categorical interpretations. This process minimized positional discrepancies and enhanced the reliability of our findings. In order to mitigate the biases caused by age differences among the different groups, such as social desirability bias, we focused on sample groups within the same age range. At the same time, we ensured that the information would not be leaked, enabling the respondents to feel more at ease in expressing their true thoughts.

To gain a deeper insight into consumers’ potential acceptance, this study carried out an in-depth analysis by using the co-occurrence analysis method of nodes. Specifically, first, the NVivo software was employed to generate an interaction matrix for the nodes. As a professional qualitative data analysis software, NVivo can efficiently process unstructured data, such as interview texts and observation records, and extract key nodes through operations like coding and classification. On this basis, the NETDRAW tool attached to UCINET was used to further generate a co-occurrence matrix, and a co-occurrence graph was produced. UCINET excels at handling relational data and accurately calculating node co-occurrence frequencies. Using the network data files generated by UCINET, NETDRAW graphically visualizes network structures. By adjusting parameters like node size and link thickness, it converts abstract co-occurrence matrices into intuitive diagrams. This series of operations are interlocked, providing intuitive and powerful data support for the subsequent research, enabling the information related to consumers’ potential acceptance to be presented in a clear and visual manner, which helps to more accurately grasp the underlying laws and trends.

## 3. Results

### 3.1. Content Analysis of the Focus Group Interviews

We encoded all the interview data and displayed the different nodes using a word cloud, as shown in Figure 1.

A visual representation of the most frequent discussion nodes among the participants is shown in Figure 1. The words with the highest frequency of occurrence were as follows: “price” (82), “meat” (73), “safety” (67), “artificial” (62), “nutrition” (58), “plant” (47), and “texture” (41). This indicates that the participants were particularly concerned with the raw materials, pricing, safety, sensory attributes, and nutritional aspects of meat substitutes during the interviews. In addition to these high-frequency nodes, several secondary nodes also emerged frequently: “try” (30), “ethical” (29), “environment” (28), “protein” (28), “taste” (25), “culture” (24), “recommend” (23), and “regulatory” (20). These terms may suggest that the participants were focused on the benefits of meat substitutes and their willingness to accept these alternatives. Furthermore, third-tier high-frequency nodes included “name” (19), “health” (17), “high technique” (15), “supervision” (15), “maturity” (14), “bean” (13), “publicize” (13), “livelihood” (13), “label” (12), “media” (11), “monopoly” (10), and “traceability” (10). These nodes may primarily represent the participants’ suggestions and expectations regarding the future development of meat substitutes.

In Figure 2, by summarizing the frequency count in the words cloud, we can see a bar distribution chart of the specific number of words. Different keywords are sorted in descending order of their frequency of occurrence, allowing us to focus on the keywords located at the top of the ranking, which will be beneficial to us in solving the current issues that hinder the acceptance of meat substitutes. For the keywords with lower rankings, these may be crucial potential influencing factors, such as pollution and ecology. As the livestock industry, climate change, and energy depletion further weaken the environment, these keywords are likely to become mainstream in the coming years.

Figure 3 summarizes the distribution of keywords concerning influencing factors and acceptance. We can draw some important information from this graph. For example, in developed cities like Beijing and Shanghai, people tend to focus on safety; this may be because food safety has become a highlight topic in China’s developed regions. The focus on food safety may lead to the outcome in Figure 3, where key words such as regulation, traceable, and label appeared in responses from residents of developed cities. However, in underdeveloped cities, people tend to focus on price, possibly due to their low income and the need to save expenses; a possible outcome of this is that people may pay less attention to qualities like taste and maturity. This accords with the statistics in Figure 3.

### 3.2. Consumers’ Knowledge of Meat Substitutes

Although most interviewees reported prior awareness of meat substitutes, their understanding of specific classifications and production processes varied. Participants from developed regions demonstrated greater familiarity with meat substitutes, frequently using terms such as “plant” and “cultured” during the interviews. These participants generally associated meat substitutes with plant proteins or animal stem cells. In contrast, participants from inland or rural areas often equated meat substitutes with legumes like tofu, as evidenced by the higher frequency of the term “bean” in these regions, as shown in Figure 1. Participants reported encountering meat substitute products through various channels, including online shopping platforms, grocery stores, and social media platforms such as TikTok. Some also gained knowledge about meat substitutes through online academic engagements, including lectures by prominent scholars like Schengen Fan, former Director General of the International Food Policy Research Institute (IFPRI), and participation in food technology competitions. For example, one participant noted, “I have previously participated in a subject knowledge competition about food technology, where I learned some information about plant-based meat and cultured meat” (Group 1, female, 26 years old).

### 3.3. Consumer Perception of Meat Substitutes

Participants’ perceptions of meat substitutes exhibited considerable variability, with some expressing a relatively positive outlook, while others adopted a more skeptical or cautious stance.

#### 3.3.1. Trust

Ecological environment protection and resource conservation

Some participants who supported meat substitutes incorporated environmental and ecological concerns into their perspectives. Figure 1 depicts several words such as “environment”, “ecology”, “pollution”, “resource”, and “water”, which may reflect these underlying concerns. During the interviews, participants underscored China’s rapidly growing population and the increasing demand for meat products. They highlighted the potential environmental impacts of this trend, including anticipated rises in greenhouse gas emissions and groundwater pollution due to the expansion of ruminant animal populations. Additionally, participants expressed concerns about the exacerbation of grassland desertification and disruption of the ecological balance resulting from the rapid growth in livestock numbers. Given China’s ambitious goals to achieve carbon peak by 2035 and carbon neutrality by 2050, participants viewed the meat substitute industry as a promising solution to mitigate future environmental and ecological pressures. They argued that adopting meat substitutes could reduce the strain on natural resources and support environmental sustainability.

2.Animal welfare

Words such as “livestock”, “ethical”, and “exploitation”, as depicted in Figure 1, may suggest that participants highlighted the positive effects of meat substitutes on animal protection and welfare during the discussions. Supporters of meat substitutes argue that these products could reduce the need for extensive animal slaughter. This is particularly relevant to the widespread use of plant-based protein alternatives, which are seen as a means to prevent the deprivation of animals’ inherent right to survival. As one participant articulated, “Cows, sheep, and chickens deserve humane treatment similar to that given to companion animals like dogs and cats. The ethical inconsistency in our treatment of pets versus livestock highlights the need to transition to meat substitutes to reduce the slaughter of these sentient beings” (participant from Group 6, female, 27 years old).

3.Cleanliness and nutritional diversity

The presence of terms such as “health”, “nutrition”, “cholesterol”, and “infect” in Figure 1 suggests that the participants viewed meat substitutes as beneficial for maintaining nutritional balance and reducing the risk of infectious diseases associated with traditional animal husbandry. Several participants highlighted the health drawbacks of conventional meat products, noting their high fat and cholesterol content, which can contribute to conditions such as hyperlipidemia and hypertension. In contrast, meat substitutes offer a customizable nutritional profile that can be tailored to individual dietary needs, addressing deficiencies in essential micronutrients such as vitamins and minerals. Additionally, the consumption of meat substitutes is associated with the reduced intake of trans fats, making them a favorable option for weight management and overall physical health.

Traditional livestock farming practices face significant health risks due to diseases such as mad cow disease, avian influenza, and swine fever. The rise of zoonotic diseases has posed serious challenges to human health in recent years. Meat substitutes produced in controlled factory or laboratory settings offer a safeguard against the transmission of animal-borne diseases to humans. As one participant noted, “Several years ago, media reports on African swine fever caused considerable concern. Despite stringent government measures, the disease’s high mortality rate led me to avoid pork during that period. The growing meat substitute industry provides relief from the anxiety associated with infectious animal diseases like swine fever” (participant from Group 2, female, 24 years old).

4.Famine alleviation

The term “famine”, as depicted in Figure 1, indicates that fewer participants perceived meat substitutes as a potential solution to hunger issues in impoverished regions. While China has made significant progress in eradicating absolute poverty and in addressing basic food and clothing needs, famine remains a persistent concern in various parts of the world, including in Africa, West Asia, and the Caribbean. Meat substitutes are seen as a timely intervention to address food shortages caused by unforeseen crises such as extreme weather events or outbreaks of animal diseases. Recent advancements in the food supply system, including the development of safe and reliable plant-based protein alternatives and cultured meat, offer promising supplementary options within the meat market. These innovations are viewed as potential food sources that could help mitigate future food security challenges.

#### 3.3.2. Negative Perceptions of Meat Substitutes

During the interviews, a subset of participants expressed dissenting views regarding meat substitutes. Their skepticism primarily focused on concerns about food safety, manufacturing techniques, potential impacts on traditional livestock husbandry, and industry monopolization. Participants who held negative perceptions articulated their objections based on the following issues:Food safety issues

The terms “safety”, “health”, “maturity”, “additives”, and “neophobia”, as shown in Figure 1, reflect the participants’ concerns about the safety of meat substitutes. Some participants view meat substitutes as a relatively new category of food in the Chinese market, characterized by complex manufacturing processes. Concerns are particularly pronounced regarding the use of emerging technologies, such as those employed in the production of animal cell-based meat alternatives, which may increase the risk of contamination during production. Additionally, there is apprehension about maintaining taste and freshness, leading companies to use additives such as enzymes, thickeners, emulsifiers, and preservatives. Some stakeholders worry that these substances could pose health risks with prolonged consumption. Sceptics also draw comparisons to genetically modified foods, highlighting potential long-term health implications of regularly consuming meat substitutes. They argue that while the immediate adverse effects might be minimal, a comprehensive assessment of the long-term effects requires extended observation. One participant expressed this skepticism, stating, “I perceive meat substitutes as similar to genetically modified foods. I will refrain from consuming them until sustained research confirms their safety for human health” (Group 3, male, 36 years old). Thus, the prevailing uncertainty about the safety of meat substitutes underscores the challenge of establishing their safety within the current discourse.

2.Laws and regulations

In Figure 1, the terms “maturity”, “regulatory”, and “supervision” indicate the participants’ concerns regarding current manufacturing practices, regulatory technologies, and the adequacy of existing laws. Participants emphasized the need for robust legal frameworks to ensure food safety and to protect consumer rights in the context of emerging food technologies, particularly artificial meat. They argue that comprehensive legal structures are necessary to address the complex challenges and opportunities associated with these innovations. Currently, domestic laws and regulations governing artificial meat are still in their early stages and lack the thoroughness needed to effectively oversee this rapidly developing industry. As a result, deficiencies in regulatory technology pose inherent risks, potentially leading to various food safety issues.

3.Unnatural taste and texture

The terms “taste”, “texture”, “unnatural”, “irregular”, and “disgusting” in Figure 1 reflect consumers’ skepticism and cautious attitudes toward the authenticity of meat substitutes. Survey respondents generally expressed concerns about the flavor profile of these products, which they perceive as artificial and reminiscent of soy-based products. For instance, one participant shared their experience: “I previously purchased meat substitutes from a supermarket and was initially impressed by their visual similarity to real meat. However, upon consumption, I found a noticeable discrepancy between the beef-like flavor and the texture, which was more akin to crumbled tofu than authentic meat” (Group 5, female, 25 years old). Additionally, participants noted that the majority of current market offerings consist of plant-based protein substitutes, which are often priced higher than traditional meat and yet fail to achieve the authentic taste that consumers seek. Another respondent reflected on their experience: “I tried a meat substitute meal out of curiosity and found it peculiar. Although I finished it due to its high cost, I am uncertain about future purchases, as I still prefer the genuine taste of real meat” (Group 4, male, 26 years old).

4.Monopoly: welfare losses for herders and consumers

In Figure 1, the terms “monopoly”, “price”, and “livelihood” frequently appear, reflecting concerns that the emergence of meat substitutes may introduce new technologies and market monopolies, potentially impacting consumer welfare and the livelihoods of traditional livestock herders. During the interviews, participants highlighted the possibility of a market consolidation in artificial meat production, with a few major enterprises dominating the sector due to technological complexities and food safety considerations. Such monopolistic scenarios could grant these enterprises significant pricing power, which may adversely affect consumer welfare. Both price and food safety emerged as critical factors influencing consumer acceptance of meat substitutes.

China, as a major center for animal husbandry, not only meets the meat consumption needs of its population but also supports the livelihoods of a significant number of pastoralists. The expansion of large-scale production of meat substitutes has the potential to encroach upon the market share traditionally occupied by conventional meat products such as pork, lamb, beef, poultry, and seafood. This shift could present considerable challenges to the livelihoods of many pastoralists and workers engaged in traditional animal husbandry practices.

### 3.4. Consumers’ Potential Acceptance of Meat Substitutes

To gain a deeper understanding of consumers’ potential acceptance of meat substitutes, this study performed an in-depth analysis using a co-occurrence analysis of nodes, as illustrated in Figure 4.

During the focus group interviews, many participants expressed a willingness to explore meat substitutes. Figure 4 illustrates the co-occurrence analysis of nodes, where the size of the vertices represents the frequency of occurrence, and the number of edges denotes the degree of co-occurrence between nodes. Most nodes are interconnected, indicating relationships among the terms. The primary concern identified is food safety. The central part of Figure 4 highlights frequently occurring nodes, such as “price”, “safety”, “meat”, “texture”, “nutrition”, “artificial”, “try”, and “plant”. These nodes are closely related to consumers’ potential acceptance of meat substitutes. The second core circle includes nodes such as “environment”, “regulation”, “ethics”, “pollution”, “technology”, “maturity”, “vegetarianism”, “taste”, “culture”, “health”, “label”, “media”, “publicity”, “legislation”, “recommendation”, “transparency”, “traceability”, and “naming”. These nodes show a moderate correlation with consumer acceptance of meat substitutes. Additionally, there is an indirect connection between peripheral terms like “curiosity” and purchase-related terms such as “try” and “buy”. Some participants’ motivation stems from a deep curiosity about these alternatives that is driven by a desire to compare their taste with that of traditional meat products. As one participant noted, “I am inclined towards experimentation and wish to personally assess the validity of claims regarding the taste of meat substitutes. However, my interest is primarily driven by curiosity; while occasional substitution is acceptable, I am not inclined towards frequent consumption” (Group 2, female, 24 years old).

The co-occurrence analysis of high-frequency words reveals that consumer acceptance of meat substitutes is more strongly associated with personal factors such as price, safety, taste, and nutrition. In contrast, the influence of societal factors, including environmental protection, animal welfare, and resource conservation, is comparatively weaker. Liu, et al. [34] similarly identified food safety and price as critical barriers to the adoption of meat substitutes. In the interviews, participants indicated that they would be reluctant to choose meat substitutes unless their prices are at least 30% lower than those of conventional meat products. This reluctance is particularly pronounced when affordable options for traditional meat are available.

The co-occurrence analysis indicates that the keywords “try” and “buy” exhibit a closer association with “plant-based” rather than “cultured”, suggesting that consumers are more inclined to accept plant-based meat compared with cultured meat [35]. The participants in the interviews highlighted that the current cost of plant-based meat products is approximately two to three times higher than that of conventional meat. Additionally, there are concerns that the future pricing of cultured meat may be even more elevated. This disparity in cost is likely a significant factor influencing consumer preference for plant-based over cultured meat alternatives.

## 4. Discussion

The text cloud and co-occurrence analysis results both demonstrate consumers’ knowledge, perception, and acceptance of meat substitutes.

### 4.1. The Role of Personal Characteristics in Consumer Knowledge

Shaw and Mac Con Iomaire [36] observed that urban consumers are more likely to accept cultured meat as an ethical alternative compared with those residing in rural areas. This study similarly found that a significant proportion of the participants lacked a comprehensive understanding of meat substitutes, with notable exceptions being among consumers from metropolitan areas such as Beijing and Shanghai. Acceptance is also contingent upon income, education level, and age; consumers might buy cultured meat if it is much cheaper than conventional meat. There is a huge difference between the willingness to try and the willingness to eat meat substitutes [12,27]; this point is also reflected in Figure 4, where the number of people willing to try these alternatives is much larger than the number of people willing to buy them, and the connection between the two is not very strong. In this study, participants from urban centers generally exhibited higher levels of education and income and tended to be predominantly younger. which contributed to their greater familiarity with and acceptance of meat substitutes. Conversely, individuals from inland or rural regions often perceive meat substitutes merely as traditional plant-based products like tofu, reflecting a limited familiarity and acceptance. These findings suggest that factors such as geographic region, age, and educational background significantly influence consumers’ knowledge, perception, and acceptance of meat substitutes.

Our findings corroborate existing research, reinforcing the notion that age significantly influences consumer attitudes and acceptance toward meat substitutes. Younger participants exhibited a more sophisticated understanding of meat alternatives and demonstrated a more favorable disposition toward both plant-based protein and cell-cultured meat, indicating a higher propensity for acceptance. In contrast, participants from focus groups in areas such as Changsha and Linyi, which are characterized by relatively older demographics, showed a less comprehensive understanding of meat substitutes and a more skeptical attitude toward their adoption. These observations align with previous studies that highlight the impact of age on consumer perceptions and acceptance of meat substitutes [37,38].

From the perspective of participants’ educational levels, those from major cities such as Beijing and Shanghai, where education levels are generally higher, demonstrated a significantly greater familiarity with meat substitutes compared with participants from regions with lower educational attainment. This finding is consistent with prior research, which indicates that higher levels of education correlate with an increased awareness and acceptance of meat alternatives [39,40,41].

In the interviews, it was observed that women generally exhibited more caution regarding meat substitutes compared with men. This finding suggests that gender is a significant factor influencing consumer attitudes toward meat alternatives. This conclusion aligns with existing research, which has similarly noted gender differences in the perceptions and acceptance of alternative proteins [42,43,44].

### 4.2. Consumer Acceptance of Meat Substitutes

Participants showed a notable interest in trying meat substitutes, largely driven by curiosity, yet they were hesitant about committing to regular, long-term consumption. This reluctance is rooted in concerns over the relative immaturity of manufacturing technologies and potential health risks associated with sustained use, which aligns with findings from previous studies [4,43].

Weinrich, et al. [45] highlighted that animal welfare and environmental concerns are key positive factors influencing consumer acceptance of meat alternatives. Mancini, et al. [46] emphasized that animal ethics is a primary driver of the increased willingness to purchase meat substitutes. In addition, cultural factors are also significant influencers. A survey of Singaporean consumers conducted by Cheon, B. K, et al. showed that conceptual differences in food essentialism are the reasons why people choose plant-based meat. That is to say, people believe that plant-based meat has similarities to traditional meat in terms of sensory and nutritional characteristics [47]. A decade-long survey conducted in Scotland revealed that although meat alternatives are now more readily available in the current environment, food culture continues to act as a barrier to consumers’ awareness and willingness to change their diets [48]. However, our research findings indicate that factors such as food safety, price, nutrition, and the naturalness of meat substitutes outweigh the importance of environmental protection, animal ethics, and the role of addressing hunger. This conclusion is consistent with the views expressed by other scholars [26,49,50,51,52].

The participants demonstrated a marked preference for plant-based protein meat over cultured meat, a finding consistent with prior research [35,49,53,54,55,56]. This preference is largely attributed to the familiarity and perceived safety of plant-based protein sources, such as tofu and bean products, which are deeply rooted in Chinese culinary traditions. Conversely, the relatively immature manufacturing technologies and underdeveloped regulatory systems for cultured meat contribute to consumer reluctance from a food safety standpoint. Additionally, the higher manufacturing costs associated with cultured meat further diminish its appeal in comparison with plant-based options. Some participants in our study estimated that it might take between 10 to 30 years or more for Chinese consumers to fully accept cultured meat as a viable alternative.

### 4.3. Traceability Systems and Relevant Laws

Previous research highlights that consumers are willing to pay a premium for meat substitutes with comprehensive traceability throughout the entire production process, rather than those with limited single-link traceability [57]. The participants insisted that thorough health and safety inspections and stringent quality controls would be necessary before cultured meat could be marketed [13]. Our study reinforces the importance of implementing a robust traceability system for the sustainable development of meat substitutes in the Chinese market. Establishing a nationwide standardized traceability framework is crucial for enabling the seamless tracking of meat substitutes from source to destination. Central to this effort is the provision of clear and informative labeling. Labels are essential for consumers to understand product quality and attributes, which directly influence their purchasing decisions and consumption patterns [58,59,60]. Research has shown that participants from Norway and Germany tend to prefer products with less processing and want to know about the processing procedures [61]. Transparent labeling ensures that consumers have access to information about ingredients and production methods, promoting market transparency and encouraging regulatory compliance among manufacturers. This, in turn, enhances product quality. Additionally, prioritizing food safety is critical. Technological advancements in food production, including those for meat substitutes, must be subjected to rigorous scientific evaluation and regulation to maintain safety standards and nutritional integrity. This approach ensures that novel products are both safe and nutritious, thereby protecting consumer health and building trust in meat substitutes.

### 4.4. Naming of Meat Substitutes

The distinction between plant-based meat and cultured meat is well established in the United States and European countries, but in the EU, cultured meat is not considered meat from a legal point of view. According to the EU regulation, it is a novel food, not meat [62]. This category encompasses various terms such as artificial meat, in vitro meat, animal-cell meat, clean meat, laboratory meat, cultured meat, and cell-based meat [53]. In contrast, in Chinese media and academic discourse, both plant-based meat and meat cultivated from animal cells are often grouped under the broad term “artificial meat” [27]. During the focus group interviews conducted in China, the participants demonstrated a higher level of acceptance for plant-based protein meat compared with cultured meat. This preference appears to be influenced by the conflation of plant-based and cultured meat under the “artificial meat” label, which may lead to negative perceptions. The term “artificial meat” carries strong technological connotations, often evoking a sense of artificiality and detachment from traditional food sources. This sentiment is consistent with research by Min, which indicates that the term “artificial meat” is less readily embraced by consumers compared with terms like “cultured meat” or “clean meat” [2]. The findings underscore the importance of precise nomenclature in the Chinese market. Scholars argue that the terminology used to describe meat substitutes significantly impacts consumer acceptance and perception [63]. Research suggests that Chinese consumers may be more receptive to terms like “cultured meat” and “cell-based meat” rather than “artificial meat” [64]. Clear and accurate naming is crucial to avoid confusion and to effectively communicate the nature and benefits of the different types of meat substitutes. This distinction can help align consumer expectations with the product’s characteristics and improve the acceptance and understanding of meat substitutes.

### 4.5. Limitations and Future Research

This study acknowledges several limitations that may impact the generalizability of its findings. First, the selection of administrative regions and participants was restricted, and there was no separate analysis of vegetarian or vegan populations. Given China’s vast land area and large population, constraints related to funding and time necessitated a limited number of interview locations and participants. Consequently, these factors may affect the representativeness of the sample and the applicability of the results to broader contexts. Future research should aim to address these limitations by including a more diverse range of locations and participant demographics, as well as by considering different dietary preferences. Such efforts will enhance the understanding of consumer behavior and preferences regarding meat substitutes and support the formulation of more comprehensive and informed policies and strategies in this evolving field.

This study primarily investigates the perception and acceptance of plant-based meat and cultured meat among Chinese consumers. With plant-based protein products increasingly entering the Chinese market, it is crucial for future research to further examine Chinese consumers’ consumption behavior in relation to these plant-based alternatives. It is important to note that meat substitutes encompass a broader range of products beyond plant-based meat and cultured meat, including plant-based seafood, cell-based seafood, and precision-fermented dairy products [65]. Future research should therefore extend its focus to include these diverse categories, exploring Chinese consumers’ knowledge, perceptions, and acceptance of such products. This will provide a more comprehensive understanding of consumer attitudes toward meat substitutes and contribute to the development of targeted strategies within this evolving market.

## 5. Conclusions

In conclusion, this study explored the knowledge, perceptions, and potential acceptance of meat substitutes among Chinese consumers, specifically focusing on plant-based and cultured meat. The research was conducted through seven focus group interviews across seven cities situated in five provincial-level administrative regions of China—Beijing, Shanghai, Hunan, Shandong, and Xinjiang. The analysis yielded several significant insights into consumer attitudes and acceptance of these alternatives.

The findings indicate that the majority of consumers have a limited understanding of meat substitutes, especially cultured meat, with only a small proportion having encountered information through platforms such as TikTok or academic lectures. Nevertheless, the participants recognized the potential advantages of meat substitutes, including environmental sustainability, improvements to animal welfare, and nutritional benefits. Concurrently, concerns were raised regarding food additives, the maturity of manufacturing technologies, sensory qualities, pricing, and regulatory frameworks.

Consumers demonstrated a willingness to explore meat substitutes that was driven by curiosity; however, broad adoption will be contingent upon advancements in manufacturing technology and improved regulatory frameworks to ensure safety and quality standards. Additionally, participants indicated that meat substitutes are unlikely to replace traditional meat products in the near future. Instead, these alternatives are expected to serve as complementary options that address the diversity in consumer preferences.

To facilitate the integration of meat substitutes into mainstream consumption, participants emphasized the necessity of prioritizing food safety through the implementation of stringent regulatory measures and the establishment of a comprehensive traceability system. Drawing on current research, it is advisable for practitioners to prioritize implementing stringent regulatory measures and traceability systems while advancing the promotion and awareness of meat substitutes through the lenses of food safety and sustainable development. The government should popularize knowledge about meat substitutes and enact regulatory laws. Enterprises are expected to establish a traceability system and to disclose detailed production process information. Consumers, on their part, should improve their knowledge base and exercise independent judgment regarding the acceptance of meat substitutes.

## Figures and Tables

**Figure 1 foods-14-01446-f001:**
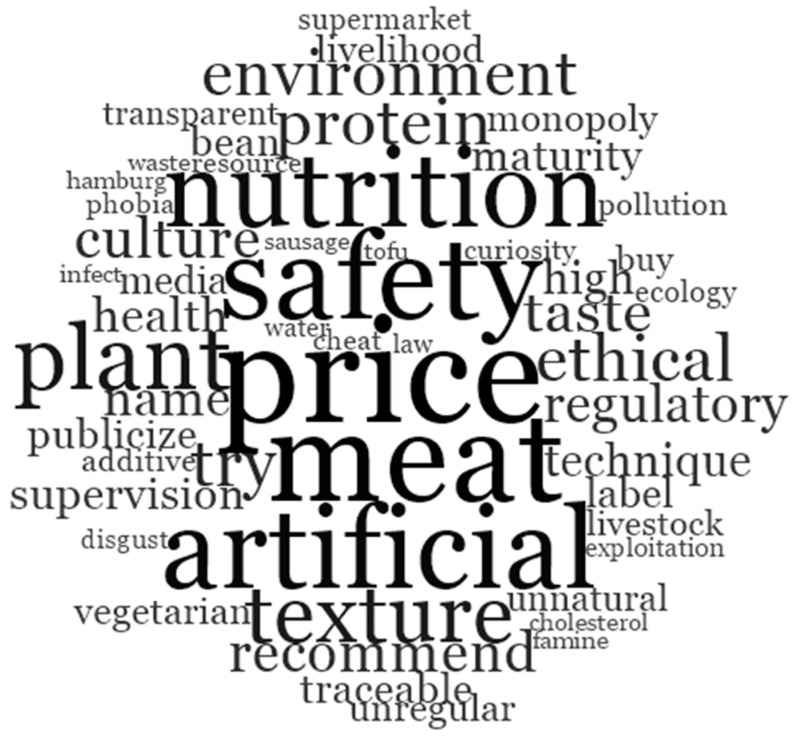
Word cloud map of participants’ responses in the focus group interviews.

**Figure 2 foods-14-01446-f002:**
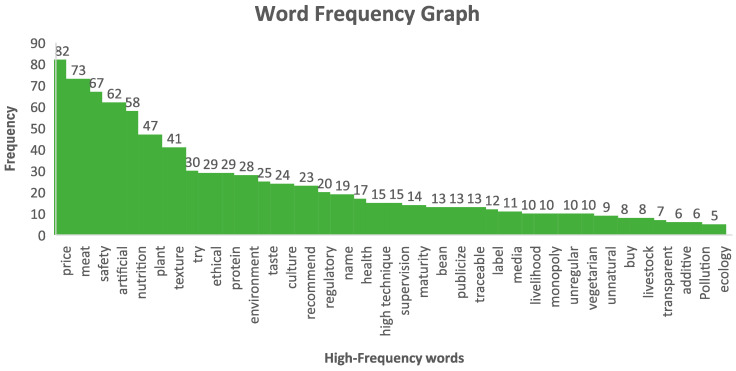
Summary of word frequencies from participants’ responses in the focus group interviews.

**Figure 3 foods-14-01446-f003:**
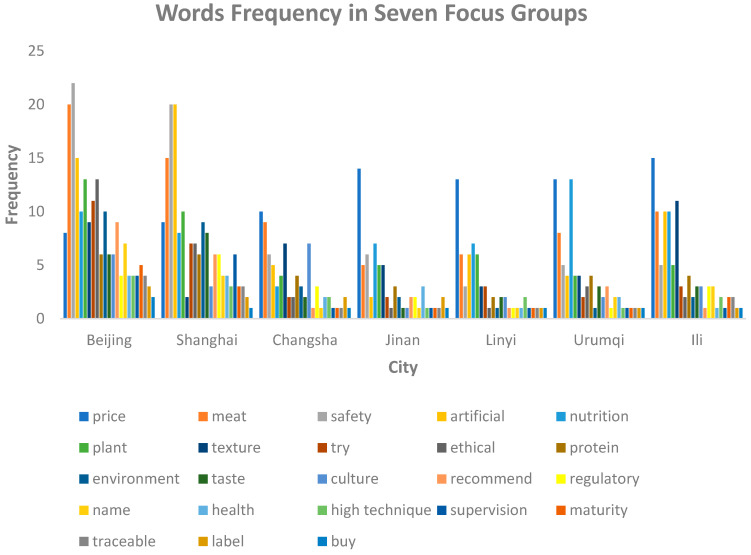
Summary of keyword frequencies for participants from each focus group interview.

**Figure 4 foods-14-01446-f004:**
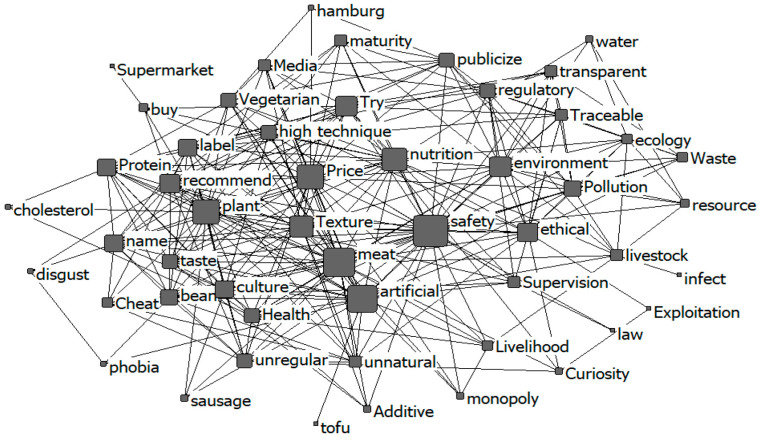
Co-occurrence analysis of nodes.

**Table 1 foods-14-01446-t001:** Description of the study participants.

Focus Group Location	Beijing	Shanghai	Changsha	Jinan	Linyi	Urumqi	Ili
Session number	1	2	3	4	5	6	7
Participant number	8	8	7	8	8	12	8
Participant code	S1	S2	S3	S4	S5	S6	S7
North/South	North	South	South	North	North	North	North
East/Center/West	East	East	Center	East	East	West	West
Age	20–30	20–30	40–60	20–30	30–45	20–30	20–30
Gender ^1^	5M3F	4M4F	3M4F	6M2F	4M4F	4M8F	6M2F
Personal income ^2^	5000–10,000	5000–8000	6000–10,000	3000–8000	2000–4000	5000–8000	3000–5000
Educational background ^3^	MD, BD	MD, BD	BD, HS	BD, HS	JMS, HS	MD, BD	BD, CD

^1^ M = male; F = female. ^2^ Yuan (China’s currency) USD 1 = CNY 7.240 (20 April 2024). ^3^ MD = master’s degree; BD = bachelor’s degree; CD = college degree; HS = high school; JMS = junior school.

**Table 2 foods-14-01446-t002:** Focus group interview outline.

Topics	Specific Issues
Knowledge of meat substitutes	1. Do you know about meat substitutes? If you know, through what channels did you learn about meat substitutes?
Perception of meat substitutes	2. What are the benefits of meat substitutes such as plant-based protein meat and cultured meat? 3. What are the potential risks of meat substitutes such as plant-based protein meat and cultured meat?
Potential acceptance of meat substitutes	4. Would you try to eat meat substitutes? Why or why not? 5. Would you consume and purchase meat substitutes regularly in the future? Why? 6. What kind of meat substitute would you willing to accept? Plant-based protein meat or animal cell-cultured meat? Why? 7. In what aspects do meat substitutes need to be improved to increase consumer acceptance of them?

## Data Availability

The original contributions presented in the study are included in the article; further inquiries can be directed to the corresponding author.

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
