# Peer review of "Exploring Chinese Consumers’ Perception and Potential Acceptance of Cell-Cultured Meat and Plant-Based Meat: A Focus Group Study and Content Analysis"

_foods, 2025, doi:10.3390/foods14091446_

Round 1

Reviewer 1 Report

Comments and Suggestions for Authors

The manuscript is well-written and presents a timely and important contribution to the understanding of Chinese consumers' perceptions and potential acceptance of cell-cultured meat and plant-based meat. The qualitative approach adopted by the authors offers valuable insights into a relatively underexplored area, particularly within the Chinese context.

I would recommend only minor revisions, as outlined below:

  1. To enhance the introduction, the authors could more introduce a theoretical framework to guide the interpretation of perceptions and acceptance.
  2. The methodological approach is appropriate. However, some aspects lack detail. Clarify how saturation was achieved and how biases (e.g., social desirability or age difference within different groups) were mitigated. 
  3. The discussion mentions traceability and legislation, but the implications are underdeveloped. The authors could state policy recommendations for regulators, food industry stakeholders, and consumer education platforms.

Overall, the manuscript offers an insightful and well-executed exploration of consumer attitudes toward meat substitutes. I thank the authors for their work and look forward to seeing the final version.

Author Response

Comments 1:To enhance the introduction, the authors could more introduce a theoretical framework to guide the interpretation of perceptions and acceptance.

Response 1:Thank you very much for your advice. We have already cited the literature in the main text to establish the connection between perception and acceptance (Line 75). Please check it.  

Comments 2:The methodological approach is appropriate. However, some aspects lack detail. Clarify how saturation was achieved and how biases (e.g., social desirability or age difference within different groups) were mitigated.   

Response 2:Thank you for your affirmation of our method. For your constructive suggestions. We have clarified how the data reached saturation at Line 230 and how to mitigate various biases at Line 239. Please review it.  

Comments 3:The discussion mentions traceability and legislation, but the implications are underdeveloped. The authors could state policy recommendations for regulators, food industry stakeholders, and consumer education platforms.  

Response 3:Thank you for your suggestions. In the conclusion part, we have put forward suggestions respectively for the government, enterprises, and consumer education platforms (Line 661). Please check it.    

Reviewer 2 Report

Comments and Suggestions for Authors

General comments

 This manuscript explore chinese consumers' perception and potential acceptance on cell-cultured meat and plant-based meat by the qualitative methodolgy of focus group and content analysis. The study is of interest to the food industry. It is well written, methodology is adequately described and give a panorama of the perceptions of chinesse consumers about new trends in food processing. A few changes and clarifications can improve the quality of this manuscript before its acceptance. Specific comments were given hereinafter.

 Specific comments

Materials and methods Add info about the software used, they are not well mentioned see lines 191 -192, 219-220.   Results I suggest to the authors to create graphs showing: the frequency of the words mentioned in the cloud map and the trends obtained for each group in order to easily visualize the responses.   Discussion The authors must compare the results obtained from chinesse people with the results obtained in other countries. For this task, they can improve the bibliography search. As an example, the following manuscripts can be cited and references list . -Bobby K. Cheon, Yan Fen Tan, Ciarán G. Forde, Food essentialism is associated with perceptions of plant-based meat alternatives possessing properties of meat-based products, Food Quality and Preference, Volume 123, 2025,https://doi.org/10.1016/j.foodqual.2024.105328 -Scheibenzuber S, Pucci E, Presenti O, Serafini G, Nobili C, Zoani C, Duta DE,Mihai AL, Criveanu-Stamatie GD, Belc N, Falch E, Rustad T and Rychlik M (2025). Consumers acceptance of new food ingredients from the food industry’s by-products—a focus group study. Front. Nutr. 12:1509833. doi: 10.3389/fnut.2025.1509833 -Emily Cleland, David McBey, Vitri Darlene, Benjamin J.J. McCormick, Jennie I. Macdiarmid, Still eating like there's no tomorrow? A qualitative study to revisit attitudes and awareness around sustainable diets after 10 years,Appetite,Volume 206,2025,107799, https://doi.org/10.1016/j.appet.2024.107799. References list must be improved adding similar studies performed in other countries   Conclusions Propose strategies to improve perception of cell-cultured meat and plant-based meat by consumers and suggest specific future research directions

Author Response

Comment 1:Materials and methods Add info about the software used, they are not well mentioned see lines 191 -192, 219-220.

Response 1:Thank you very much for your advice. We have already introduced the information about the software in Lines 247-249 and Lines 251-254. Please check it.

Comment 2:Results I suggest to the authors to create graphs showing: the frequency of the words mentioned in the cloud map and the trends obtained for each group in order to easily visualize the responses.  

Response 2:Thank you for your suggestions on this paper. We have created charts to display the occurrence frequencies of the words mentioned in the word cloud (Line 279) and the trends obtained for each group (Line 290). Please review.

Comment 3:Discussion The authors must compare the results obtained from chinesse people with the results obtained in other countries. For this task, they can improve the bibliography search. As an example, the following manuscripts can be cited and references list . -Bobby K. Cheon, Yan Fen Tan, Ciarán G. Forde, Food essentialism is associated with perceptions of plant-based meat alternatives possessing properties of meat-based products, Food Quality and Preference, Volume 123, 2025,https://doi.org/10.1016/j.foodqual.2024.105328 -Scheibenzuber S, Pucci E, Presenti O, Serafini G, Nobili C, Zoani C, Duta DE,Mihai AL, Criveanu-Stamatie GD, Belc N, Falch E, Rustad T and Rychlik M (2025). Consumers acceptance of new food ingredients from the food industry’s by-products—a focus group study. Front. Nutr. 12:1509833. doi: 10.3389/fnut.2025.1509833 -Emily Cleland, David McBey, Vitri Darlene, Benjamin J.J. McCormick, Jennie I. Macdiarmid, Still eating like there's no tomorrow? A qualitative study to revisit attitudes and awareness around sustainable diets after 10 years,Appetite,Volume 206,2025,107799, https://doi.org/10.1016/j.appet.2024.107799. References list must be improved adding similar studies performed in other countries   Conclusions Propose strategies to improve perception of cell-cultured meat and plant-based meat by consumers and suggest specific future research directions

Response 3:Thank you so much for the literature you sourced for us. We have cited these references and added a comparison of consumer acceptance across different countries(Line 544-551,577-579 ).Please kindly review.
